# GENERATING LESS CERTAIN ADVERSARIAL EXAMPLES IMPROVES ROBUST GENERALIZATION

## ABSTRACT

Recent studies have shown that deep neural networks are vulnerable to adversarial examples. Numerous defenses have been proposed to improve model robustness, among which adversarial training is most successful. In this work, we revisit the robust overfitting phenomenon. In particular, we argue that overconfident models produced during adversarial training could be a potential cause, supported by the empirical observation that the predicted labels of adversarial examples generated by models with better robust generalization ability tend to have significantly more even distributions. Based on the proposed definition of adversarial certainty, we incorporate an extragradient step in the adversarial training framework to search for models that can generate adversarially perturbed inputs with lower certainty, further improving robust generalization. Our approach is general and can be easily combined with other variants of adversarial training methods. Extensive experiments on image benchmarks demonstrate that our method effectively alleviates robust overfitting and is able to produce models with consistently improved robustness.

## 1 INTRODUCTION

Deep neural networks (DNNs) have achieved exceptional performance and been widely adopted in various applications, including computer vision (He et al., 2016), natural language processing (Devlin et al., 2019) and recommendation systems (Covington et al., 2016). However, DNNs have been shown to be vulnerable to classifying inputs, also known as adversarial examples (Szegedy et al., 2014; Goodfellow et al., 2015), crafted with imperceptible perturbations that are designed to trick the model into making wrong predictions. The prevalence of adversarial examples has raised serious concerns regarding the robustness of DNNs, especially when deployed in security-critical and safety-critical applications such as self-driving cars (Chen et al., 2015), biometric facial recognition (Komkov & Petiushko, 2021) and medical diagnosis (Finlayson et al., 2019; Ma et al., 2021).

To improve the resilience of DNNs against adversarial examples, numerous defense mechanisms have been proposed, e.g., distillation (Papernot et al., 2016), adversarial detection (Ma et al., 2018), feature denoising (Xie et al., 2019), randomized smoothing (Cohen et al., 2019) and semi-supervised methods (Alayrac et al., 2019). Among them, adversarial training (Madry et al., 2018; Zhang et al., 2019) is by far one of the most effective methods in producing adversarially robust models. However, even the state-of-the-art adversarial training methods (Croce et al., 2020; Rebuffi et al., 2021; Wang et al., 2023) cannot achieve satisfactory robustness performance on simple classification tasks like classifying CIFAR-10 images. Witnessing the empirical challenges in further improving adversarial robustness, many recent studies focus on understanding the behavior of adversarial training (Tu et al., 2019; Gao et al., 2019; Wu et al., 2020; Zhang et al., 2020; Yu et al., 2022). In particular, Rice et al. (2020) observed that the testing-time model robustness immediately increases by a large margin after the first learning rate decay but keeps decreasing afterwards during adversarial training, termed as the *robust overfitting* phenomenon. Robust overfitting has attracted a lot of attention since then, as overfitting is not an issue for standard deep learning but appears to be dominant for adversarially-trained DNNs. In addition, recognizing the fundamental cause of robust overfitting may provide us with important insights in designing better ways to produce more robust models.

This paper revisits the robust overfitting phenomenon to provide a different perspective on explaining why it happens and develop a new algorithmic solution inspired by the gained insight to enhance robust generalization. More specifically, we observe that models produced during adversarial training

tend to be overconfident in predicting the label of the training-time adversarial examples generated by the model but not for adversarially-perturbed testing samples, where the gap potentially induces robust overfitting. Inspired by this observation, we propose an extragradient step in adversarial training to search models that can generate less certain adversarial examples.

**Contributions.** By comparing the classwise distributions of predicted labels of adversarial examples generated at different epochs during adversarial training, we observe that models with better robust generalization ability exhibit more even distributions. In contrast, the final model produced by adversarial training is significantly overconfident in predicting the labels of training-time adversarial examples, resulting in robust overfitting (Figure 1). We introduce a general notion of *adversarial certainty* to capture a model's confidence in classifying adversarial examples generated by the model at a logit level (Definition 2.1), and show a similar trend that models with better robust generalization have lower adversarial certainty with respect to training examples, which motivates us to prevent adversarially-trained models from being overconfident (Section 2).

Built upon the definition, we propose a novel ***Extragradient-type method to explicitly Decrease Adversarial Certainty (EDAC)*** for adversarial training (Section 3). EDAC first finds the steepest descent direction of model weights to decrease adversarial certainty, aiming to generate less certain adversarial examples during training, then the newly generated adversarial examples with lower certainty are used to optimize model robustness. As the model learns from less certain adversarial examples, the robust generalization gap between training and testing will then be shrunk. Experiments on image benchmark datasets demonstrate that our method consistently produces more robust models when combined with various adversarial training methods, confirming the importance of generating less certain adversarial examples, and that robust overfitting is significantly mitigated with the help of our EDAC (Section 4.1). Besides, we investigate the cooperation of adversarial certainty with other helpful insights for model robustness, where our method shows further generalizability by compatibly improving them (Section 4.2). In general, these improvements empirically depict the correlation with adversarial certainty, indicating that our method helps generate less certain adversarial examples and improves the robust generalization of adversarially-trained models.

**Related Work.** Adversarial training is a promising defense framework for improving model robustness against adversarial examples (Goodfellow et al., 2015; Madry et al., 2018; Zhang et al., 2019; Wang et al., 2020; Tramèr et al., 2017; Shafahi et al., 2019; Andriushchenko & Flammarion, 2020; Wong et al., 2020; Jin et al., 2022). In particular, Goodfellow et al. (2015) proposed to adversarially train models using perturbations generated by the fast gradient sign method (FGSM). Later on, Madry et al. (2018) incorporated perturbations produced by iterative projected gradient decent (PGD) into adversarial training, which learns models with more reliable and robust performance. Other variants of adversarial training have been proposed, which typically modify the training objective but also use PGD. For instance, Zhang et al. (2019) designed TRADES, which considers the standard classification loss and encourages the decision boundary to be smooth, while Wang et al. (2020) proposed MART to emphasize the importance of misclassified examples during adversarial training.

Besides improving adversarial training, several recent works focus on understanding robust generalization and leveraging the gained insight to build more robust models (Rice et al., 2020; Stutz et al., 2021; Hwang et al., 2021; Chen et al., 2021; Yu et al., 2022; Xu et al., 2023). In particular, Rice et al. (2020) discovered that, unlike standard deep learning, robust overfitting is a dominant phenomenon for adversarially-trained DNNs that hinders robust generalization, and advocated the use of early stopping, while Wu et al. (2020) discovered that the flatness of weight loss landscape is an important factor related to robust generalization, which inspires them to adversarially perturb the model weights during adversarial training. In addition, Tack et al. (2022) proposed a consistency regularization term based on data augmentation to mitigate robust overfitting. Our work complements these methods, where we explain why overconfidence in generating adversarial examples is highly related to the robust overfitting phenomenon and illustrate how to improve robust generalization by promoting less certain perturbed inputs for adversarial training.

We are also aware of two recent works that focus on improving adversarial training with the consideration of model overconfidence (Stutz et al., 2020; Setlur et al., 2022). These works are aligned with our insight for designing EDAC but target different objectives. Specifically, Stutz et al. (2020) developed a confidence-calibrated adversarial training method that achieves better robustness against unseen attacks, while Setlur et al. (2022) proposed a regularization technique to maximize the entropy

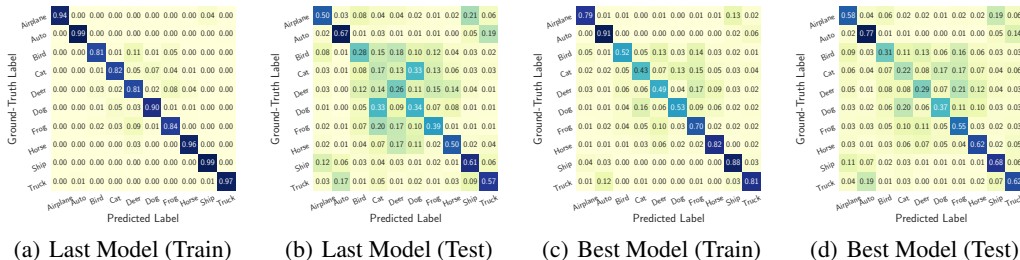

(a) Last Model (Train)     (b) Last Model (Test)     (c) Best Model (Train)     (d) Best Model (Test)

Figure 1: Heatmaps of predicted class distribution of training-time and testing-time generated adversarial examples with respect to models produced from the last epoch and the best epoch of adversarial training, where darker colors indicate larger probabilities.

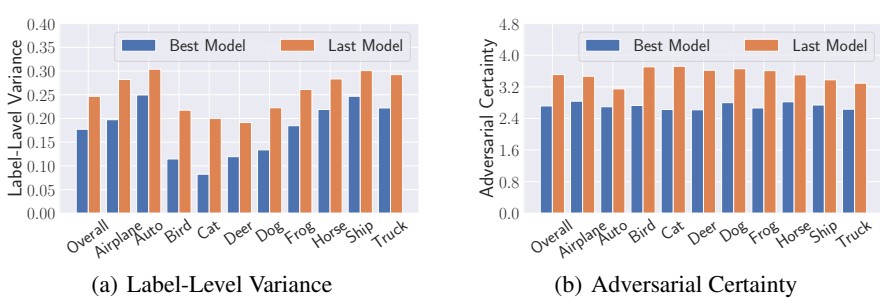

(a) Label-Level Variance        (b) Adversarial Certainty

Figure 2: Histograms of (a) label-level variances and (b) adversarial certainty of training-time adversarial examples generated by *Last Model* and *Best Model* across different ground-truth CIFAR-10 classes.

of model predictions on out-of-distribution data with larger perturbations, thus improving model accuracy on unseen examples.

## 2   ADVERSARIAL TRAINING PRODUCES OVERCONFIDENT MODELS

This section explains why robust overfitting happens from the perspective of overconfidence. Due to the space limit, we present the necessary notations and related concepts, including adversarial robustness, robust generalization and adversarial training, with detailed definitions in Appendix A.

To gain a deeper understanding of the phenomenon, we plot the heatmaps of the predicted label distribution for the adversarially-perturbed CIFAR-10 images in Figure 1, respectively generated by the model with the best robust generalization during adversarial training, denoted as *Best Model*, and the final model produced in the last epoch, denoted as *Last Model*. Here, the ground-truth label represents the underlying class of clean images and the predicted label is the class of adversarially-perturbed images generated by PGD-Attack predicted by the corresponding model. Figure 1 shows disparate characteristics between the label distributions of training- and testing-time adversarial examples with respect to the best and last models produced from adversarial training. Specifically, the testing-time predicted label distributions look much less certain than the training-time ones. Besides, in the training time, the distribution with respect to the *Last Model* mainly concentrates on the ground-truth class, whereas the *Best Model* with better robust generalization produces a more even distribution which is more similar to that of the corresponding testing samples. More experimental details and the scaled-up version of Figure 1 are depicted in Appendix B.

Moreover, we measure the variance of the class probabilities of the predicted labels, denoted as *label-level variance*, with respect to the training-time adversarial examples for each ground-truth category in Figure 2(a). In particular, we observe that the *Best Model* with better robust generalization performance exhibits a lower label-level variance than that of the *Last Model*, which is consistent with

the results illustrated in Figure 1. Even though the label-level variance best characterizes the predicted label distribution and reflects how certain the training-time generated adversarial examples are, such statistics are on a classwise distribution level, which is not easy to optimize. Therefore, we propose an alternative logit-level definition, termed as *adversarial certainty*, to capture the certainty of a model in classifying the adversarial examples generated by some attack method (e.g., PGD-Attack), where a lower value indicates lower model certainty in predicting adversarial examples:

**Definition 2.1** (Adversarial Certainty). Let $\mathcal{X}$ be the input space and $\mathcal{Y}$ be the label space. Suppose $\mu$ is the underlying distribution and $\mathcal{S}$ is a set of sampled examples. Let $\epsilon \geq 0$, $\Delta$ be the perturbation metric and $\mathcal{A}$ be some specific attack method for generating adversarial examples. For any neural network $M_\theta : \mathcal{X} \to \mathcal{Y}$, we define the *adversarial certainty* of $M_\theta$ with respect to $\mathcal{S}$ and $\mathcal{A}$ as:

$$\mathrm{AC}_\epsilon(M_\theta; \mathcal{S}, \mathcal{A}) = \frac{1}{|\mathcal{S}|} \sum_{(\boldsymbol{x},y) \in \mathcal{S}} \mathrm{Var}\left( \widetilde{M_\theta}\big[ \mathcal{A}(\boldsymbol{x}; y, M_\theta, \epsilon) \big] \right). \tag{1}$$

Here, $\widetilde{M_\theta} : \mathcal{X} \to \mathbb{R}^{|\mathcal{Y}|}$ is the mapping from the input space to the logit layer of $M_\theta$ and $\mathrm{Var}(\boldsymbol{u}) = \left( \sum_{k \in [K]} (u_k - \overline{\boldsymbol{u}})^2 / K \right)^{1/2}$ for any $\boldsymbol{u} \in \mathbb{R}^K$, where $\overline{\boldsymbol{u}}$ denotes the average of all elements in $\boldsymbol{u}$.

Different from the label-level variance, adversarial certainty is an averaged sample-wise metric, which computes the variance of the logits returned by the model $M_\theta$ for each adversarially-perturbed example $\mathcal{A}(\boldsymbol{x}; y, M_\theta, \epsilon)$. Similar to Figure 2(a), we compute the adversarial certainty of the *Best Model* and the *Last Model* for each ground-truth category, which is further visualized in Figure 2(b). Since predicted labels are decided by the corresponding class with the highest predicted probabilities, the adversarial certainty depicts a similar pattern to the label-level variance as expected.

**Our Perspective.** Based on the empirical observations demonstrated in Figures 1 and 2, we further explain why model overconfidence is not beneficial for robust generalization, which is a potential cause for robust overfitting. Adversarial training aims to learn a model that can best classify the adversarial examples generated based on the model itself. Such an objective will encourage the model to keep improving its confidence in predicting the ground-truth with respect to the adversarially-perturbed training examples, which eventually leads to producing an overconfident model. However, this overconfidence property is detrimental to robust generalization, because the model cannot generate perturbed training inputs with low enough certainty, which is different from the typical generated adversarial examples during testing time. Consequently, the model will not be able to well predict the less certain test-time adversarial examples, thus resulting in a larger robust generalization gap between training and testing time. We hypothesize that such a large gap limits adversarially robust generalization. To the best of our knowledge, this perspective is new to the field of adversarial machine learning and is intuitively aligned with the classical machine learning theory that if the testing distribution deviates more from the training distribution (in our case, the prediction certainty of adversarial examples between training and testing), standard machine learners are expected to exhibit a decreased generalization performance (in our case, the robust generalization performance).

## 3 DECREASING ADVERSARIAL CERTAINTY WITH EXTRAGRADIENT STEPS

Motivated by the findings in Section 2, we propose a novel *Extragradient-type method to explicitly Decrease Adversarial Certainty* (EDAC), which searches for models with lower adversarial certainty by involving an extragradient step (Mertikopoulos et al., 2019; Zhang & Yu, 2020) in each training iteration to improve robust generalization.[1] More specifically, EDAC aims to solve the following optimization problem to encourage models to generate less certain adversarial examples:

$$\min_{\theta \in \Theta} \frac{1}{|\mathcal{S}_{tr}|} \sum_{(\boldsymbol{x},y) \in \mathcal{S}_{tr}} \max_{\boldsymbol{x}' \in \mathcal{B}_\epsilon(\boldsymbol{x})} L\big(M_{\theta'}, \boldsymbol{x}', y\big), \text{ where } \theta' = \underset{\theta' \in \mathcal{C}(\theta)}{\mathrm{argmin}} \; \mathrm{AC}_\epsilon(M_{\theta'}; \mathcal{S}_{tr}, \mathcal{A}), \tag{2}$$

---

[1]Our work is mainly supported by intuitive explanation and empirical evidence. Regarding rigorous theoretical analysis, we think it would be a non-trivial task as both the training and testing distributions of adversarial examples are dynamically changed during adversarial training, which is a clear difference from the standard supervised machine learning regime. We view it as an interesting future work and plan to investigate it under simplified settings as the next step, e.g., optimization with a single iteration on a robust-overfitting model.

where $\mathcal{S}_{tr}$ is the clean training dataset, $\mathcal{A}$ denotes a specific attack method (e.g., the PGD-Attack $\mathcal{A}_{\text{pgd}}$), and $\mathcal{C}(\theta)$ represents the feasible search region for $\theta'$. Here, $\theta'$ can be understood as a function of $\theta$ which we aim to optimize in Equation 2. We remark that imposing the constraint of $\mathcal{C}(\theta)$ is necessary since adversarial certainty should be optimized towards that of testing examples to increase the training-testing similarity of predicted label distributions, whereas unconstrained optimization will cause $\theta'$ deviating too much from the initial $\theta$, inducing a negative impact on robust generalization (see Figure 3(b) for more results and discussions regarding the design choice of the constraint set).

Directly solving the min-max-min problem specified in Equation 2 is challenging due to the non-convex nature of the optimization and the implicit definition of $\mathcal{C}(\theta)$, thus we resort to gradient-based methods for an approximate solver. To be more specific, we take the $(t+1)$-th iteration of adversarial training as an example to illustrate our design of EDAC. Given a set of clean training examples $\mathcal{S}_{tr}$, a specific attack method $\mathcal{A}$, and a classification model $M_\theta$, our EDAC method can be formulated as a two-step optimization for the $(t+1)$-th iteration:

$$
\begin{aligned}
\theta_{t+0.5} &= \theta_t - \eta \cdot \nabla_\theta \text{AC}_\epsilon(M_\theta; \mathcal{S}_{tr}, \mathcal{A})\Big|_{\theta=\theta_t}, \\
\theta_{t+1} &= \theta_{t+0.5} - \gamma \cdot \nabla_\theta L_{\text{rob}}(M_\theta; \mathcal{S}_{tr}, \mathcal{A})\Big|_{\theta=\theta_{t+0.5}},
\end{aligned}
\tag{3}
$$

where $\eta > 0$ and $\gamma > 0$ represent the step sizes of the two optimization steps, $\text{AC}_\epsilon(M_\theta; \mathcal{S}_{tr}, \mathcal{A})$ denotes the adversarial certainty of $M_\theta$ with respect to $\mathcal{S}_{tr}$ and $\mathcal{A}$ in Definition 2.1, and $L_{\text{rob}}(M_\theta; \mathcal{S}_{tr}, \mathcal{A})$ can be roughly understood as a robust loss except the inner maximization is approximated using some attack method $\mathcal{A}$. The first step in Equation 3 optimizes the adversarial certainty, where it changes the model parameters $\theta_t$ in a direction that decreases the model's adversarial certainty the most, whereas the second step in Equation 3 optimizes the model's ability in distinguishing adversarial examples generated by the model itself as used in standard adversarial training.

As a result, the adversarial certainty of $M_{\theta_{t+0.5}}$ will be decreased after the first optimization step, i.e., $\text{AC}_\epsilon(M_{\theta_{t+0.5}}; \mathcal{S}_{tr}, \mathcal{A}) < \text{AC}_\epsilon(M_{\theta_t}; \mathcal{S}_{tr}, \mathcal{A})$. According to the insights presented in Section 2, the model is expected to gain a better robust generalization ability after conducting the second optimization step on less certain adversarial examples generated based on $M_{\theta_{t+0.5}}$. Thus, as adversarial training proceeds, such decreased adversarial certainty and improved robust generalization ability will be iteratively learned to the final model, which mitigates the undesirable robust overfitting and improves model robustness. In each iteration, the extragradient step optimizes adversarial certainty to help find less certain adversarial examples corresponding to the current model status. This is why EDAC could work even if decreasing cross-entropy loss in the previous iteration and decreasing adversarial certainty in the current iteration are in different directions, i.e., $M_{\theta_{t+1}}$ will enjoy the newly optimized adversarial certainty, compared with $M_{\theta_t}$.

**Correlation Analysis.** Since our work aims to improve robust generalization by decreasing adversarial certainty, it is natural to ask:

*Does lower adversarial certainty imply better robust generalization?*

Recall that in Equation 2, $\mathcal{C}(\theta)$ defines the feasible region for optimizing adversarial certainty. Therefore, the answer should intuitively be positive within this region, i.e., decreasing adversarial certainty will increase test robust accuracy. To better answer this question with empirical evidence, we conduct a correlation analysis between adversarial certainty and robust generalization. The results are illustrated in Figure 3(b). Specifically, we use an AT-trained model as the starting point, from which the heatmaps (Figure 1) are derived. Then, we separately update one more epoch on the model by EDAC using different step sizes (from 0.1 to 2.0) in extragradient steps to decrease adversarial certainty. Afterward, we measure the training-time adversarial certainty (i.e., the blue bars) and robust test accuracy (i.e., the orange curve) of the result models. As expected, adversarial certainty keeps decreasing with the increase in step size. Meanwhile, the model robustness first keeps improving, but it decreases when the step size is larger than 1.3. These results indicate that when the step size is appropriately selected, the optimized model parameters are still within the feasible search region, wherein lower training-time adversarial certainty corresponds to higher robustness. However, when the model is out of the feasible search region, decreasing adversarial certainty will not help improve robust generalization anymore.

**Why Extragradient Steps?** Our algorithmic design of utilizing extragradient steps is inspired by the existing literature on min-max optimization (Daskalakis & Panageas, 2018; Diakonikolas et al., 2021;

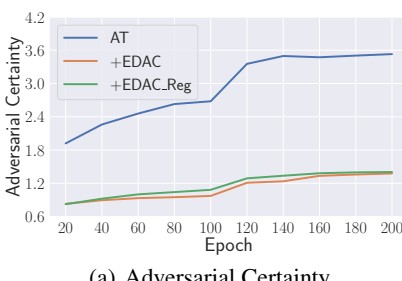
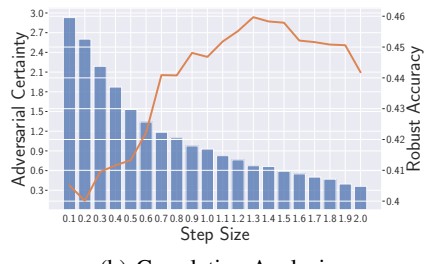

(a) Adversarial Certainty            (b) Correlation Analysis

Figure 3: (a) Training curves of adversarial certainty for different methods. (b) Correlation analysis between adversarial certainty and robust accuracy, where x-axis is the step size used in extra gradient, blue bars are training-time adversarial certainty, and the orange curve denotes robust test accuracy.

Hsieh et al., 2021; Pethick et al., 2022). In particular, these related works illustrate that standard first-order methods, such as stochastic gradient descent/ascent (SGDA), can converge to recurrent orbits that do not contain any critical point of the underlying min-max problem, regardless of the basic convex-concave regime or general non-convex/non-concave problems. Extragradient-type methods have been studied as a first-order alternative to SGDA that can avoid such failure modes and have proven to be a principled way to find local solutions (e.g., a stationary point) to the underlying min-max problems under various settings. This sheds light on the potential of using extragradient steps to solve min-max problems more effectively. Nevertheless, our proposed method that optimizes for adversarial robustness is slightly different from the standard extragradient method typically studied in the optimization literature, because the extra gradient is computed with respect to adversarial certainty instead of cross-entropy loss. Still, it shares a similar insight of escaping the failure mode of the standard first-order min-max solver with the help of extragradient steps. As will be demonstrated in Section 4, the introduced extragradient step for optimizing adversarial certainty is essential to obtain a stable and improved robustness performance across various benchmarks and different settings, compared with other optimization schemes (e.g., adding an extra regularizer on adversarial certainty).

**Application Scope.** Our method can be summarized as a two-step iterative optimization framework: $\theta_t \longrightarrow \theta_{t+0.5} \longrightarrow \theta_{t+1}$. We remark that this framework provides a generic way to explicitly optimize a specified property (e.g., adversarial certainty) in the first step, and then to take advantage of the newly generated training samples with a better property to find a more robust model in the second step. Intuitively, this framework is generally applicable to any adversarial defense that requires training-time adversarial examples generated based on the to-be-updated model, because the optimization of adversarial robustness relies on the generated adversarial examples, i.e., training samples with updated properties would benefit model robustness. In Section 4, we will examine the compatibility of our EDAC method with several PGD-based methods, including adversarial training and its variants.

## 4 EXPERIMENTS

In this section, we evaluate the effectiveness of our EDAC method in improving robust generalization with respect to different model architecture PreActResNet-18 (PRN18) and WideResNet-34 (WRN34) under the threat model of $\ell_\infty$ trained by several benchmark adversarial defenses, i.e., AT (Madry et al., 2018), TRADES (Zhang et al., 2019) and MART (Wang et al., 2020) on CIFAR-10 in Section 4.1. Then, we evaluate the compatibility of EDAC with AWP (Wu et al., 2020) and Consistency (Tack et al., 2022) in Section 4.2. In addition, to examine the generalizability to other settings, we provide results of EDAC on more benchmark datasets, including CIFAR-10, CIFAR-100 and SVHN, and $\ell_2$ norm-bounded perturbations in Appendix D. All experimental details are depicted in Appendix C.

### 4.1 MAIN RESULTS

**Results on CIFAR-10.** We evaluate the robust generalization of our proposed EDAC on CIFAR-10, a commonly-used image benchmark. The results are depicted in Table 1. We can see that, EDAC

Table 1: Testing-time adversarial robustness (%) with/without EDAC on CIFAR-10 under $\ell_\infty$ threat model across different model architectures and adversarial training methods.

| Architecture | Method | Clean | PGD-20 | PGD-100 | CW$_\infty$ | AutoAttack |
|---|---|---|---|---|---|---|
| PRN18 | AT | 82.88 (82.68) | 41.51 (49.23) | 40.96 (48.92) | 41.61 (48.07) | 39.66 (45.71) |
| | **+ EDAC** | **84.64 (83.55)** | **45.55 (52.20)** | **44.94 (51.87)** | **44.55 (50.05)** | **42.78 (48.20)** |
| | TRADES | 82.10 (81.33) | 47.44 (51.65) | 46.95 (51.42) | 46.64 (49.18) | 44.99 (48.06) |
| | **+ EDAC** | **83.18 (82.80)** | **49.32 (52.90)** | **48.81 (52.67)** | **48.30 (50.11)** | **46.40 (48.96)** |
| | MART | 80.85 (78.27) | 50.23 (52.28) | 49.71 (52.13) | 46.88 (47.83) | 44.68 (46.01) |
| | **+ EDAC** | **81.12 (79.37)** | **52.38 (53.25)** | **52.04 (53.14)** | **48.97 (49.25)** | **47.24 (47.69)** |
| WRN34 | AT | 86.47 (**85.86**) | 47.25 (55.31) | 46.73 (55.00) | 47.85 (54.04) | 45.84 (51.94) |
| | **+ EDAC** | **86.48** (85.10) | **52.02 (57.93)** | **51.69 (57.68)** | **51.51 (54.98)** | **49.75 (53.33)** |
| | TRADES | 86.01 (84.74) | 49.66 (53.72) | 48.44 (53.60) | 48.56 (52.35) | 46.53 (51.31) |
| | **+ EDAC** | **86.75 (85.18)** | **53.70 (55.86)** | **53.09 (55.59)** | **52.73 (53.72)** | **50.50 (52.42)** |
| | MART | 83.11 (81.31) | 48.93 (53.87) | 48.31 (53.68) | 46.32 (49.65) | 44.89 (48.00) |
| | **+ EDAC** | **84.69 (83.23)** | **52.00 (55.57)** | **51.32 (55.22)** | **49.50 (51.43)** | **47.65 (49.92)** |

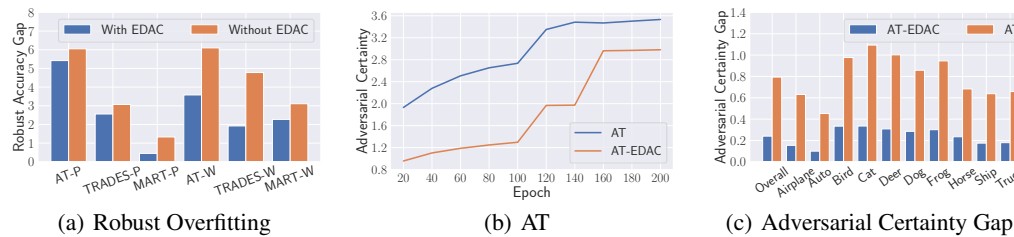

(a) Robust Overfitting          (b) AT          (c) Adversarial Certainty Gap

Figure 4: The visualized results evaluating robust overfitting and adversarial certainty, where "-P" and "-W" in the x-axis of Figure 4(a) represent PRN18 and WRN34 respectively.

significantly enhances testing-time adversarial robustness across different adversarial attacks. These results show the effectiveness of EDAC, indicating the significance of less certain adversarial examples for robust generalization. Besides, we investigate the influence of EDAC on robust overfitting which is an attention-worthy phenomenon. Specifically, Figure 4(a) evaluates the gap of testing-time adversarial robustness between the best and the last models. We can observe that EDAC consistently mitigates robust overfitting across different settings. These results indicate that decreasing adversarial certainty could mitigate the degradation of robust generalization induced by robust overfitting.

Moreover, we find that WRN34 suffers from more severe robust overfitting in adversarial training, however, could benefit more mitigation by using our EDAC method. This suggests that WRN34 is superior to PRN18 in terms of robust generalization with the help of EDAC. Besides adversarial robustness, the clean accuracy for testing images is also worth attention since they measure the standard generalization ability of the model. As expected, EDAC consistently improves the clean test accuracy in all cases as shown in the *Clean* column of Table 1. This promotion shows that EDAC also could help models gain better robustness against unseen clean images even by learning from adversarial examples, which also benefits from the improvement of generalization.

**EDAC Improves Adversarial Certainty.** Since EDAC has been empirically shown effective to improve robust generalization, in this part, we want to demonstrate this is because EDAC could decrease adversarial certainty. To this end, we train a sequence of models by AT and TRADES for 200 epochs, and by MART for 120 epochs, respectively. And for every 20 epochs in this sequence, we update the same intermediate model by one further epoch using the original adversarial defenses with and without the help of EDAC, respectively. Then, we measure the adversarial certainty of all further-updated models following Equation 1. In Figure 4(b), we compare AT with AT-EDAC and observe that – consistent with our findings in Section 2 and design purpose explained in Section 3 – EDAC could indeed help models generate less certain adversarial examples, which would bring better robust generalization. The cases of TRADES and MART are measured in Figure 7(a) and Figure 7(b)

Table 2: Testing-time adversarial robustness (%) of AT with/without EDAC/EDAC_Reg on CIFAR-10 and PRN18 under $\ell_\infty$ threat model.

| Architecture | Method | Clean | PGD-20 | PGD-100 | CW$_\infty$ | AutoAttack |
|---|---|---|---|---|---|---|
| PRN18 | AT | 82.88 (82.68) | 41.51 (49.23) | 40.96 (48.92) | 41.61 (48.07) | 39.66 (45.71) |
| | **+ EDAC** | **84.64 (83.55)** | **45.55 (52.20)** | **44.94 (51.87)** | **44.55 (50.05)** | **42.78 (48.20)** |
| | **+ EDAC_Reg** | 83.78 (83.54) | 45.39 (50.86) | 44.87 (50.49) | 44.18 (48.96) | 42.41 (47.02) |

Table 3: Testing-time adversarial robustness (%) of AWP and Consistency with/without EDAC on CIFAR-10 and PRN18 under $\ell_\infty$ threat model.

| Method | Clean | PGD-20 | PGD-100 | CW$_\infty$ | AutoAttack |
|---|---|---|---|---|---|
| AT-AWP | 83.76 (82.37) | 52.99 (54.04) | 52.71 (53.89) | 51.07 (51.22) | 48.75 (49.33) |
| **+ EDAC** | **84.07 (82.67)** | **54.55 (55.17)** | **54.30 (55.00)** | **51.76 (52.03)** | **49.80 (49.96)** |
| AT-Consistency | 85.28 (84.66) | 55.43 (56.72) | 55.16 (56.46) | 50.81 (51.13) | 48.08 (48.48) |
| **+ EDAC** | **85.36 (85.17)** | **56.65 (57.19)** | **56.31 (56.90)** | **51.29 (51.72)** | **49.00 (49.46)** |

respectively in Appendix D, which show similar trends. Moreover, we measure the adversarial certainty gap between the best model and the last model on AT and AT-EDAC in Figure 4(c). We find that the adversarial certainty gap of AT-EDAC is significantly smaller, which indicates closer adversarial robustness of the best model and the last model. This result explains why EDAC could mitigate robust overfitting by decreasing adversarial certainty in extragradient steps.

**Alternative Regularization Method.** As we only conceptually illustrate involving extragradient steps for EDAC in Section 3, this section will empirically demonstrate the necessity of this decision by comparing EDAC with decreasing adversarial certainty without extragradient steps. Specifically, we construct a regularization term to decrease the adversarial certainty, and add it to robust loss. Then, similar to adversarial training, in each iteration, we only conduct a single step of optimization according to the regularized robust loss, which is termed as EDAC_Reg. Due to the space limit, we only depict the evaluation of a PRN18 model on CIFAR-10 in Table 2, while the full results on SVHN, CIFAR-10 and CIFAR-100 datasets with both PRN18 and WRN34 models are shown in Table 4 in Appendix D.

We can see that EDAC_Reg could improve the robust generalization of AT due to the regularization to decrease adversarial certainty, but EDAC brings better and more stable improvements, which indicates the advantage of involving extragradient steps. These results comply with our discussion in Section 3 that existing literature on min-max optimization shows the potential of using extragradient-type methods to avoid limit cycles with respect to standard gradient descent/ascent approaches. In addition, we measure the adversarial certainty of a sequence of models trained by AT, EDAC and EDAC_Reg, respectively, in Figure 3(a). We could observe that EDAC gains the lowest adversarial certainty (even only with a slight gap over EDAC_Reg), indicating that higher robust generalization corresponds to lower adversarial certainty.

## 4.2 Extension to Other Adversarial Training Methods

Previous sections show the efficacy of EDAC on adversarial training and its variants. We note that some recent works also focus on understanding robust generalization and developing methods to improve adversarial training towards more robust models, including Adversarial Weight Perturbation (Wu et al., 2020) – AWP for short – and Consistency Reguarlaization (Tack et al., 2022) – Consistency for short. However, since these methods focus on different insights to improve robust generalization, whether our proposed EDAC framework is compatible with them remains elusive. In this section, we empirically study the generalizability of our EDAC to AWP and Consistency regularization. We evaluate the case of AT in Table 3, while the full results of AT, TRADES and MART are provided in Table 5 in Appendix D.

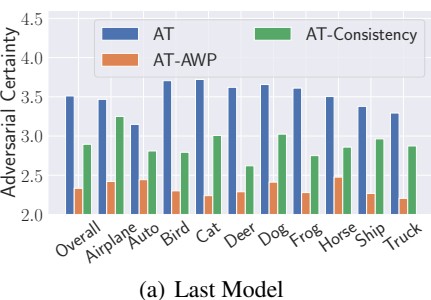
(a) Last Model

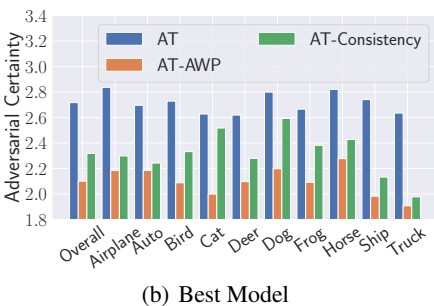
(b) Best Model

Figure 5: The visualized results on CIFAR-10 with respect to PRN18 for comparing adversarial certainty of AT, AT-AWP and AT-Consistency based on (a) last model, and (b) best model.

**Adversarial Weight Perturbation.** AWP discovered that the flatness of the weight loss landscape is an important factor related to robust generalization, while EDAC seeks less certain training-time adversarial examples. AWP shares some similarities with our work, where both EDAC and AWP find new model status by gradient-based method before optimizing model robustness, i.e., in the $(t + 1)$-th iteration, model parameters $\theta_{t+1}$ is derived from the optimization on $\theta_{t+0.5}$, instead of directly from $\theta_t$. However, there is a main difference between AWP and our work: In our work, $\theta_{t+0.5} \longrightarrow \theta_{t+1}$ is optimized on the adversarial examples generated by $\theta_{t+0.5}$, which takes advantage of the optimized adversarial certainty of $\theta_{t+0.5}$, while AWP uses the adversarial examples generated by $\theta_t$ to derive $\theta_{t+1}$. Consequently, EDAC and AWP aim at different directions to help models. In that case, we attempt to cooperate EDAC with AWP by adding another extragradient step before weight perturbation to optimize the certainty of adversarial examples, and use the updated intermediate model to generate new adversarial examples for the following AWP optimization. As shown in Table 3, EDAC could compatibly work with AWP and gain further improvements in robust generalization.

**Consistency Regularization.** The method of Tack et al. (2022) regularizes the adversarial consistency based on various data augmentations. In that case, we first use extragradient steps to update the adversarial certainty on augmented samples, and then follow the Consistency optimization. From Table 3, we can see that EDAC could cooperatively improve the robust generalization of Consistency, which shows the feasibility of EDAC in the augmentation-based domain.

In general, the improvement of EDAC on AWP and Consistency is not as significant as previous ones. We hypothesize this is because AWP and Consistency could implicitly decrease adversarial certainty when helping models toward their specified directions. Thus, when EDAC conducts the explicit optimization of adversarial certainty, it can only achieve slight improvements that are derived from less certain training samples generated in extragradient steps. To this end, we measure the adversarial certainty of AWP and Consistency on the last and the best models of AT in Figure 5. We could see that both methods of AT-AWP and AT-Consistency, which achieve better robust generalization performance than AT, also implicitly decrease the adversarial certainty. This suggests that different insights for helping model robustness would be compatible with each other to some extent, but how to coordinate them for a better unified improvement is worth exploring in the future.

# 5 CONCLUSION

We revisited the robust overfitting phenomenon of adversarial training and discovered that overfitting to training data might result from the training-time adversarial examples generated by overconfident models. Specifically, we observed that models with better robust generalization performance correspond to significantly more even predicted label distributions of training-time adversarial examples. Built upon a new notion of adversarial certainty, we proposed to involve an extragradient step to generate less certain examples by decreasing adversarial certainty, which can be combined with various attack methods for generating adversarial examples. Extensive experiments substantiated the effectiveness of our method in mitigating robust overfitting and learning more robust models.

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
