# A PRELIMINARIES

**Notation.** We use lowercase boldfaced letters to denote vectors, $\text{sgn}(\cdot)$ to denote the sign function, and $\mathbb{1}(\cdot)$ to denote the indicator function. For any vector $\boldsymbol{x}$ and finite-sample set $\mathcal{S}$, let $x_k$ be the $k$-th element of $\boldsymbol{x}$ and $|\mathcal{S}|$ be the cardinality of $\mathcal{S}$. Consider a metric space $(\mathcal{X}, \Delta)$, where $\mathcal{X}$ is the input space and $\Delta : \mathcal{X} \times \mathcal{X} \to \mathbb{R}$ is a distance metric. For any $\boldsymbol{x} \in \mathcal{X}$ and $\epsilon > 0$, let $\mathcal{B}_\epsilon(\boldsymbol{x}; \Delta) = \{\boldsymbol{x}' \in \mathcal{X} : \Delta(\boldsymbol{x}', \boldsymbol{x}) \le \epsilon\}$ be the ball centered at $\boldsymbol{x}$ with radius $\epsilon$ and metric $\Delta$, where we write $\mathcal{B}_\epsilon(\boldsymbol{x}) = \mathcal{B}_\epsilon(\boldsymbol{x}; \Delta)$ when $\Delta$ is free of context. For any set $\mathcal{C} \subseteq \mathcal{X}$, let $\Pi_\mathcal{C}(\boldsymbol{x}) = \text{argmin}_{\boldsymbol{x}' \in \mathcal{C}} \Delta(\boldsymbol{x}', \boldsymbol{x})$ be the projection of $\boldsymbol{x}$ onto $\mathcal{C}$. Let $\mu$ be a probability distribution on $\mathcal{X}$. The empirical distribution of $\mu$ based on a sample set $\mathcal{S}$ is defined as: $\hat{\mu}_\mathcal{S}(\mathcal{C}) = \sum_{\boldsymbol{x} \in \mathcal{S}} \mathbb{1}(\boldsymbol{x} \in \mathcal{C})/|\mathcal{S}|$ for any $\mathcal{C} \subseteq \mathcal{X}$.

**Adversarial Robustness.** Adversarial robustness captures the classifier's resilience to small adversarial perturbations. In particular, we work with the following definition of adversarial robustness:

**Definition A.1** (Adversarial Robustness). Let $\mathcal{X} \subseteq \mathbb{R}^n$ be input space, $\mathcal{Y}$ be label space, and $\mu$ be the underlying distribution of inputs and labels. Let $\Delta$ be a distance metric on $\mathcal{X}$ and $\epsilon > 0$. For any classifier $M_\theta : \mathcal{X} \to \mathcal{Y}$, the *adversarial robustness* of $M_\theta$ with respect to $\mu$, $\epsilon$ and $\Delta$ is defined as:

$$\text{Rob}_\epsilon(M_\theta; \mu) = 1 - \Pr_{(\boldsymbol{x}, y) \sim \mu} \left[ \exists \, \boldsymbol{x}' \in \mathcal{B}_\epsilon(\boldsymbol{x}) \text{ s.t. } M_\theta(\boldsymbol{x}') \ne y \right]. \tag{4}$$

When $\epsilon = 0$, $\text{Rob}_0(M_\theta; \mu)$ is equivalent to the clean accuracy of $M_\theta$. In practice, the probability density function of the underlying distribution $\mu$ is typically unknown. Instead, we only have access to a set of test examples $\mathcal{S}_{te}$ sampled from $\mu$, thus a classifier's adversarial robustness is estimated by replacing $\mu$ in Equation 4 with its empirical counterpart based on $\mathcal{S}_{te}$. To be more specific, the testing-time adversarial robustness of $M_\theta$ with respect to $\mathcal{S}_{te}$, $\epsilon$ and $\Delta$ is given by:

$$\text{Rob}_\epsilon(M_\theta; \hat{\mu}_{\mathcal{S}_{te}}) = 1 - \frac{1}{|\mathcal{S}_{te}|} \sum_{(\boldsymbol{x}, y) \in \mathcal{S}_{te}} \max_{\boldsymbol{x}' \in \mathcal{B}_\epsilon(\boldsymbol{x})} \mathbb{1}\left(M_\theta(\boldsymbol{x}') \ne y\right), \tag{5}$$

where $\hat{\mu}_{\mathcal{S}_{te}}$ denotes the empirical measure of $\mu$ based on $\mathcal{S}_{te}$. We remark that *robust generalization*, the main subject of this study, captures how well a model can classify adversarially-perturbed inputs that are not used for training, which is essentially the testing-time adversarial robustness $\text{Rob}_\epsilon(M_\theta; \hat{\mu}_{\mathcal{S}_{te}})$. In this work, we focus on the $\ell_p$-norm distances as the perturbation metric $\Delta$, since they are most widely-used in existing literature on adversarial examples. Although $\ell_p$ distances may not best reflect the human-perceptual similarity (Sharif et al., 2018) and perturbation metrics beyond $\ell_p$-norm such as geometrically transformed adversarial examples (Kanbak et al., 2018; Xiao et al., 2018) were also considered in literature, there is still a significant amount of interest in understanding and improving model robustness against $\ell_p$ perturbations. We hope that our insights gained from $\ell_p$ perturbations will shed light on how to learn better robust models for more realistic adversaries.

**Adversarial Training.** Among all the existing defenses against adversarial examples, *adversarial training* (Madry et al., 2018; Zhang et al., 2019; Carmon et al., 2019) is most promising in producing robust models. Given a set of training examples $\mathcal{S}_{tr}$ sampled from $\mu$, adversarial training aims to solve the following min-max optimization problem:

$$\min_{\theta \in \Theta} L_{\text{rob}}(M_\theta; \mathcal{S}_{tr}), \text{ where } L_{\text{rob}}(M_\theta; \mathcal{S}_{tr}) = \frac{1}{|\mathcal{S}_{tr}|} \sum_{(\boldsymbol{x}, y) \in \mathcal{S}_{tr}} \max_{\boldsymbol{x}' \in \mathcal{B}_\epsilon(\boldsymbol{x})} L\left(M_\theta, \boldsymbol{x}', y\right). \tag{6}$$

Here, $\Theta$ denotes the set of model parameters, and $L$ is typically set as a convex surrogate loss such that $L(M_\theta, \boldsymbol{x}, y)$ is an upper bound on the 0-1 loss $\mathbb{1}(M_\theta(\boldsymbol{x}) \ne y)$ for any $(\boldsymbol{x}, y)$. For instance, $L$ is set as the cross-entropy loss in vanilla adversarial training (Madry et al., 2018), whereas the combination of a cross-entropy loss for clean data and a regularization term for robustness is used in TRADES (Zhang et al., 2019). In theory, if $\mathcal{S}_{tr}$ well captures the underlying distribution $\mu$ and the robust loss $L_{\text{rob}}(M_\theta; \mathcal{S}_{tr})$ is sufficiently small, then $M_\theta$ is guaranteed to achieve high adversarial robustness $\text{Rob}_\epsilon(M_\theta; \mu)$.

However, directly solving the min-max optimization problem 6 for non-convex models such as neural networks is challenging. It is typical to resort to some good heuristic algorithm to approximately solve the problem, especially for the inner maximization problem. In particular, (Madry et al., 2018)

proposed to alternatively solve the inner maximization using an iterative projected gradient descent method (PGD-Attack) and solve the outer minimization using SGD, which is regarded as the go-to approach in the research community. We further explain its underlying mechanism below. For any intermediate model $M_\theta$ produced during adversarial training, the *PGD-Attack* updates the (perturbed) inputs according to the following update rule:

$$\boldsymbol{x}_{s+1} = \Pi_{\mathcal{B}_\epsilon(\boldsymbol{x})}\big(\boldsymbol{x}_s + \alpha \cdot \mathrm{sgn}(\nabla_{\boldsymbol{x}_s} L(M_\theta, \boldsymbol{x_s}, y))\big) \text{ for any } (\boldsymbol{x}, y) \text{ and } s \in \{0, 1, \ldots, S-1\}, \quad (7)$$

where $\boldsymbol{x}_0 = \boldsymbol{x}$, $\alpha > 0$ denotes the step size and $S$ denotes the total number of iterations. For the ease of presentation, we use $\mathcal{A}_{\mathrm{pgd}}$ to denote the PGD-Attack such that for any example $(\boldsymbol{x}, y)$ and classifier $M_\theta$, it generates $\boldsymbol{x}' = \boldsymbol{x}_S = \mathcal{A}_{\mathrm{pgd}}(\boldsymbol{x}; y, M_\theta, \epsilon)$ based on the update rule 7. After generating the perturbed input for each example in a training batch, the model parameter $\theta$ is then updated by a single SGD step with respect to $L(M_\theta, \boldsymbol{x}', y)$ for the outer minimization problem in Equation 6.

# B EXPERIMENTAL DETAILS OF SECTION 2

In this section, we provide the scaled-up version of Figure 1 for better presentation in Figure 6, and then explain the experimental details for producing the heatmaps and the histograms illustrated in Section 2.

Given a model $M_\theta$ (e.g., *Best Model* and *Last Model*) and a set of examples $\mathcal{S}$ sampled from the underlying distribution $\mu$ (e.g., CIFAR-10 training and testing datasets), adversarial examples are generated by the PGD-Attack within the perturbation ball $\mathcal{B}_\epsilon(\boldsymbol{x})$ centered at $\boldsymbol{x}$ with radius $\epsilon = 8/255$ under the $\ell_\infty$ threat model, which follows the settings of generating training samples in Section 4, e.g., the PGD-Attack is iteratively conducted by 10 steps with the step size of $2/255$. We record the predicted labels of the generated adversarial examples with respect to each model, and then plot the classwise label distributions as heatmaps in Figure 1.

In general, let HM be the $m \times m$ matrix representing the heatmap, where $\mathcal{Y} = \{1, 2, \ldots, m\}$ denotes the label space. For any $j, k \in \mathcal{Y}$, the $(j, k)$-th entry of HM with respect to $M_\theta$ and $\mathcal{S}$ is defined as:

$$\mathrm{HM}_{j,k} = \frac{\Big|\big\{(\boldsymbol{x}, y) \in \mathcal{S} : y = j \text{ and } M_\theta\big(\mathcal{A}_{\mathrm{pgd}}(\boldsymbol{x}; y, M_\theta, \epsilon)\big) = k\big\}\Big|}{\Big|\big\{(\boldsymbol{x}, y) \in \mathcal{S} : y = j\big\}\Big|}, \quad (8)$$

where $\mathcal{A}_{\mathrm{pgd}}$ is the PGD-Attack defined by the update rule 7. More specifically, for any $(\boldsymbol{x}, y) \in \mathcal{S}$, we construct the corresponding adversarial example by the PGD-Attack, i.e., $\boldsymbol{x}' = \mathcal{A}_{\mathrm{pgd}}(\boldsymbol{x}; y, M_\theta, \epsilon)$. Then, we measure the predicted label $\hat{y} = M_\theta(\boldsymbol{x}')$. In that case, for the given training data, we could construct (*ground-truth*, *predicted*) label pairs, simply denoted by $\{(y, \hat{y})\}$. Afterward, we first cluster $\{(y, \hat{y})\}$ separately by the ground-truth label, e.g., the subset of ground-truth label $j$ includes all pairs such that $y = j$ (denoted by $\{(y, \hat{y})\}_j$), which corresponds to the rows of heatmaps. Further, for each subset, we group it into sub-subsets separately by the predicted labels, e.g., $\{(y, \hat{y})\}_{j,k}$ contains all pairs in $\{(y, \hat{y})\}_j$ such that $\hat{y} = k$. Consequently, the number of adversarial examples of the ground truth label $j$ is calculated as:

$$|\{(y, \hat{y})\}_j| = \Big|\big\{(\boldsymbol{x}, y) \in \mathcal{S} : y = j\big\}\Big|.$$

Meanwhile, the number of adversarial examples of ground truth label $j$ but predicted as label $k$ is measured as:

$$|\{(y, \hat{y})\}_{j,k}| = \Big|\big\{(\boldsymbol{x}, y) \in \mathcal{S} : y = j \text{ and } M_\theta\big(\mathcal{A}_{\mathrm{pgd}}(\boldsymbol{x}; y, M_\theta, \epsilon)\big) = k\big\}\Big|,$$

where $\hat{y} = M_\theta\big(\mathcal{A}_{\mathrm{pgd}}(\boldsymbol{x}; y, M_\theta, \epsilon)\big)$. Finally, we compute the $(j, k)$-th entry of the heatmap $\mathrm{HM}_{j,k}$ as the ratio of $|\{(y, \hat{y})\}_{j,k}|$ to $|\{(y, \hat{y})\}_j|$, i.e., Equation 8.

Moreover, following the same settings, we plot the corresponding label-level variance and adversarial certainty in Figure 2. Specifically, we first measure the label-level variance of the training-time adversarial examples of the last model (Figure 1(a)) and the best model (Figure 1(c)) conditioned on

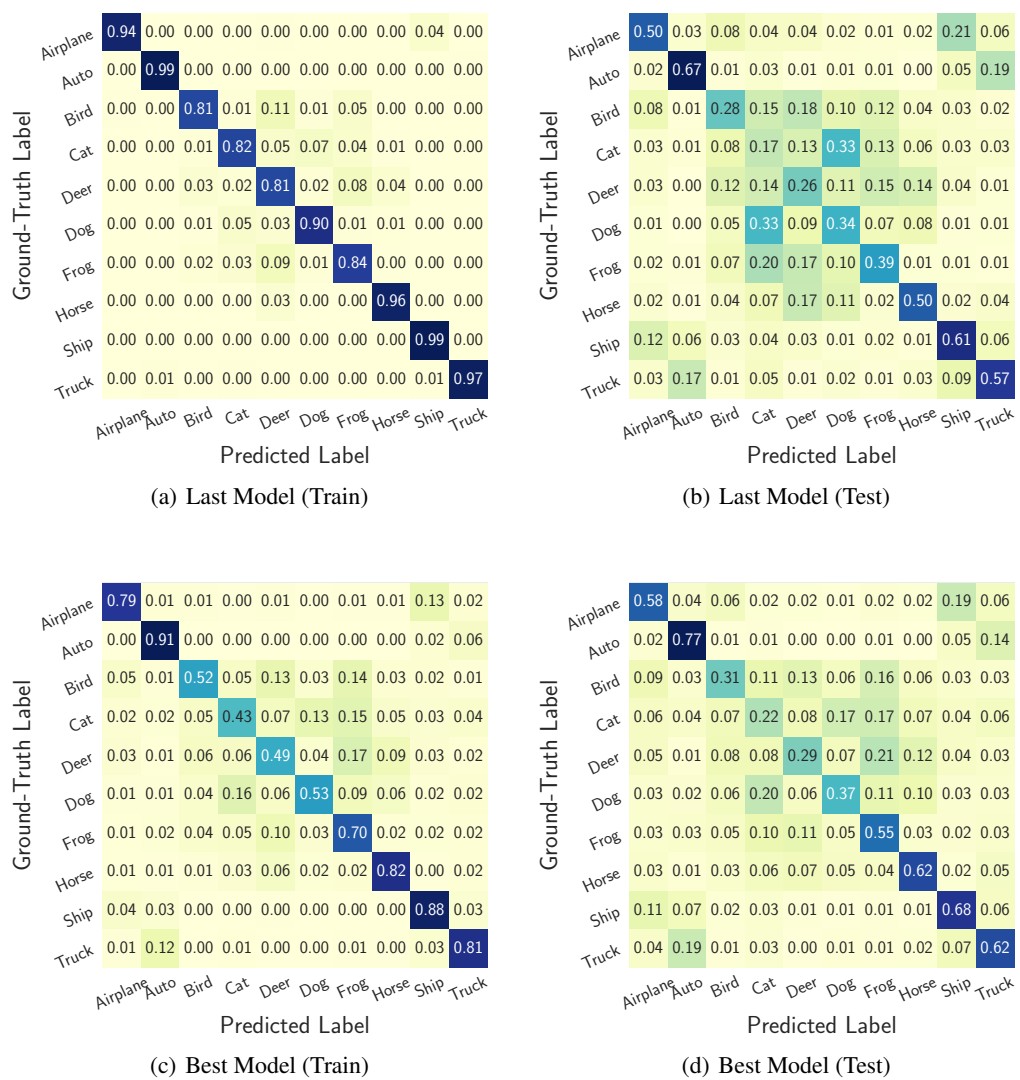

Figure 6: Heatmaps of predicted class distribution of training-time and testing-time generated adversarial examples with respect to models produced from the last epoch and the best epoch of adversarial training, where darker colors indicate larger probabilities.

the ground-truth label, as shown in Figure 2(a). Taking the ground-truth label $j$ as an example, the label-level variance can be formulated as:

$$\mathrm{Var}_j^{(\mathrm{label})} = \sqrt{\frac{1}{|\mathcal{Y}|} \sum_{k \in \mathcal{Y}} (\mathrm{HM}_{j,k} - \overline{\mathrm{HM}}_j)^2},$$

where $\overline{\mathrm{HM}}_j$ averages all $\mathrm{HM}_{j,k}$ with different $k$, and $\mathcal{Y} = \{1, 2, ..., m\}$ is the label space.

In addition, according to Equation 1 defined in Section 2, we measure the adversarial certainty of the last and the best models, as illustrated in Figure 2(b), with respect to the predicted logits of all the adversarial examples conditioned on the ground-truth label.

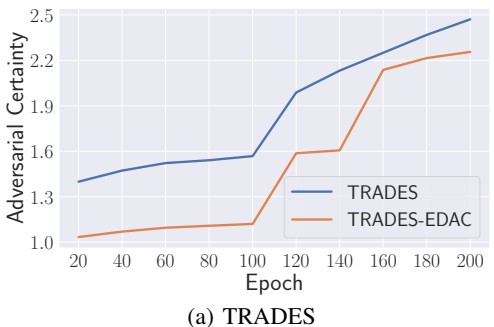
(a) TRADES

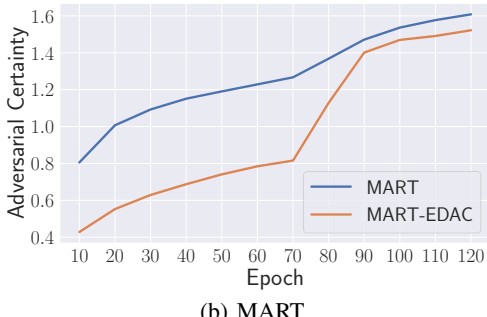
(b) MART

Figure 7: The visualized results evaluating adversarial certainty based on PRN18 of TRADES and MART using CIFAR-10 at different epochs.

Table 4: Testing-time adversarial robustness (%) of AT with/without EDAC/EDAC_Reg under $\ell_\infty$ threat model across different model architectures and benchmark datasets.

| Dataset | Architecture | Method | Clean | PGD-20 | PGD-100 | CW$_\infty$ | AutoAttack |
|---|---|---|---|---|---|---|---|
| SVHN | PRN18 | AT | 89.63 (88.64) | 42.25 (51.00) | 41.37 (50.30) | 42.84 (48.19) | 39.52 (46.02) |
| | | + EDAC | 90.58 (89.63) | **45.86 (54.42)** | **43.92 (53.78)** | **43.75 (50.15)** | 40.68 (**48.23**) |
| | | + EDAC_Reg | **90.65 (90.21)** | 45.39 (53.06) | 43.77 (52.28) | 43.66 (49.64) | **41.10** (47.39) |
| | WRN34 | AT | 91.51 (89.72) | 46.81 (53.43) | 44.94 (52.77) | 45.76 (50.43) | 41.71 (49.50) |
| | | + EDAC | 91.26 (91.83) | 60.42 (**67.95**) | 56.71 (**64.85**) | 56.98 (**65.09**) | 42.33 (**50.42**) |
| | | + EDAC_Reg | **91.76 (92.13)** | **62.19** (65.96) | **59.54** (63.68) | **60.05** (63.87) | **42.46** (49.95) |
| CIFAR-10 | PRN18 | AT | 82.88 (82.68) | 41.51 (49.23) | 40.96 (48.92) | 41.61 (48.07) | 39.66 (45.71) |
| | | + EDAC | **84.64 (83.55)** | **45.55 (52.20)** | **44.94 (51.87)** | **44.55 (50.05)** | **42.78 (48.20)** |
| | | + EDAC_Reg | 83.78 (83.54) | 45.39 (50.86) | 44.87 (50.49) | 44.18 (48.96) | 42.41 (47.02) |
| | WRN34 | AT | 86.47 (**85.86**) | 47.25 (55.31) | 46.73 (55.00) | 47.85 (54.04) | 45.84 (51.94) |
| | | + EDAC | **86.48** (85.10) | **52.02 (57.93)** | **51.69 (57.68)** | **51.51 (54.98)** | **49.75 (53.33)** |
| | | + EDAC_Reg | 85.69 (76.89) | 48.81 (48.91) | 47.54 (48.86) | 47.55 (45.98) | 44.24 (44.99) |
| CIFAR-100 | PRN18 | AT | 54.58 (53.64) | 20.29 (27.80) | 20.00 (27.66) | 20.18 (25.40) | 18.52 (23.45) |
| | | + EDAC | **54.85 (55.01)** | **22.46** (27.73) | **22.19** (27.48) | **21.11** (25.37) | 19.09 (**23.95**) |
| | | + EDAC_Reg | 54.67 (53.11) | 21.78 (**28.86**) | 21.50 (**28.70**) | 20.56 (**26.00**) | **19.29** (23.40) |
| | WRN34 | AT | 57.23 (54.45) | 25.64 (30.30) | 25.38 (29.97) | 24.09 (27.57) | 22.76 (25.46) |
| | | + EDAC | **58.15** (58.04) | **26.08 (31.55)** | **25.89 (31.43)** | **24.77 (29.19)** | **23.66 (27.08)** |
| | | + EDAC_Reg | 57.57 (**58.34**) | 24.46 (30.97) | 24.13 (30.89) | 24.04 (28.92) | 22.68 (26.71) |

## C EXPERIMENTAL DETAILS OF SECTION 4

For the empirical evaluation in Section 4, we set $\epsilon = 8/255$, and train the model for 200 epochs using SGD with the momentum of 0.9. Besides, the initial learning rate is 0.1, and is divided by 10 at the $100^{th}$ epoch and at the $150^{th}$ epoch. And the adversarial attack used in training is PGD-10 with a step size of $2/255$, while we utilize the commonly-used attack benchmarks of PGD-20 (Madry et al., 2018), PGD-100 (Madry et al., 2018), CW$_\infty$ (Carlini & Wagner, 2017) and AutoAttack (Croce & Hein, 2020) in testing. In addition, we also measure the *Clean* performance to investigate the influence on clean images. Regarding other hyperparameters, we follow the settings described in their original papers. In all cases, we evaluate the performance of the last (best) model in terms of testing-time robust accuracy.

## D ADDITIONAL RESULTS

**EDAC Improves Adversarial Certainty.** Recall that we compare the adversarial certainty of AT and AT-EDAC in Figure 4(b) (Section 4.1). We further depict the influence of EDAC on other adversarial training methods, including TRADES and MART, in Figure 7. In particular, we observe similar trends as in Figure 4(b), i.e., our EDAC could indeed help gain better adversarial certainty.

Table 5: Testing-time adversarial robustness (%) of AWP and Consistency with/without EDAC on CIFAR-10 and PRN18 under $\ell_\infty$ threat model.

| Method | Clean | PGD-20 | PGD-100 | CW$_\infty$ | AutoAttack |
|---|---|---|---|---|---|
| AT-AWP | 83.76 (82.37) | 52.99 (54.04) | 52.71 (53.89) | 51.07 (51.22) | 48.75 (49.33) |
| **+ EDAC** | **84.07** (**82.67**) | **54.55** (**55.17**) | **54.30** (**55.00**) | **51.76** (**52.03**) | **49.80** (**49.96**) |
| TRADES-AWP | 81.46 (81.28) | 52.71 (53.69) | 52.54 (53.55) | 50.37 (50.61) | 49.54 (49.92) |
| **+ EDAC** | **82.69** (**82.85**) | **54.06** (**54.68**) | **53.80** (**54.49**) | **51.44** (**51.53**) | **50.51** (**50.63**) |
| MART-AWP | 78.13 (77.27) | 53.26 (53.62) | 53.06 (52.58) | 49.05 (48.39) | 46.53 (47.01) |
| **+ EDAC** | **80.03** (**78.65**) | **54.79** (**55.16**) | **54.67** (**54.93**) | **49.58** (**49.14**) | **47.47** (**47.73**) |
| AT-Consistency | 85.28 (84.66) | 55.43 (56.72) | 55.16 (56.46) | 50.81 (51.13) | 48.08 (48.48) |
| **+ EDAC** | **85.36** (**85.17**) | **56.65** (**57.19**) | **56.31** (**56.90**) | **51.29** (**51.72**) | **49.00** (**49.46**) |
| TRADES-Consistency | 83.68 (83.51) | 53.00 (53.06) | 52.78 (52.79) | 48.85 (48.89) | 47.75 (47.77) |
| **+ EDAC** | **84.78** (**84.73**) | **53.73** (**53.96**) | **53.48** (**53.72**) | **49.37** (**49.41**) | **48.15** (**48.19**) |
| MART-Consistency | 78.21 (78.11) | 56.33 (56.85) | 56.31 (56.81) | 47.33 (47.47) | 45.53 (45.73) |
| **+ EDAC** | **81.91** (**81.35**) | **58.59** (**58.76**) | **58.29** (**58.56**) | **50.08** (**50.21**) | **48.28** (**48.59**) |

Table 6: Testing-time adversarial robustness (%) of AT with/without EDAC on PreActResNet-18 under $\ell_2$ threat model against PGD-20 across different benchmark datasets.

| Method | SVHN | | CIFAR-10 | | CIFAR-100 | |
|---|---|---|---|---|---|---|
| | Best | Last | Best | Last | Best | Last |
| AT (Madry et al. 2018) | 66.45 | 63.20 | 66.02 | 65.18 | 39.23 | 35.68 |
| **+ EDAC** | **69.11** | **67.44** | **69.10** | **67.37** | **40.75** | **36.32** |

**Other Benchmark Datasets.** We extend the evaluation of EDAC to more benchmark datasets, including SVHN (Netzer et al., 2011), CIFAR-10 (Krizhevsky & Hinton, 2009) and CIFAR-100 (Krizhevsky & Hinton, 2009), on the adversarial robustness. Table 4 shows that EDAC consistently improves robust generalization across different datasets. For instance, surprisingly, EDAC could increase the adversarial robustness by 14.52% for the WRN34 best model of AT against PGD-20. These improvements further demonstrate the effectiveness and generalizability of EDAC.

**Comparisons With Alternative Regularization Method.** As shown in Table 4, compared with EDAC_Reg, involving extragradient steps could bring better and more stable improvements. In some cases, EDAC_Reg is even harmful to robust generalization, e.g., when a WideResNet-34-10 model is trained on CIFAR-10.

**Extension to Other Adversarial Training Methods.** In Section 4.2, we depict the generalizability of EDAC to AT-AWP and AT-Consistency. We now provide the full results of the cases of AT, TRADE and MART in Table 5. Similarly, EDAC shows consistent improvements in extensive cases, which indicates the compatibility of adversarial certainty with other insights for improving robust generalization.

$\ell_2$ **Norm-Bounded Perturbation.** In the above evaluation, we focus on the $\ell_\infty$ norm-bounded perturbations. Meanwhile, the $\ell_2$ norm is also a prevalent perturbation setting in adversarial training. Thus, in Table 6, we evaluate our method under the $\ell_2$ threat model. Similarly, EDAC depicts consistent improvements in adversarial robustness on best and last epochs and different benchmark datasets, which shows the efficacy of EDAC against adversarial attacks with $\ell_2$ perturbations.