# OpenReview forum: "Generating Less Certain Adversarial Examples Improves Robust Generalization"
_ICLR.cc/2024/Conference — ICLR 2024 Conference Withdrawn Submission_

### Official Review · Reviewer_4Bie · 2023-10-29

**Soundness:** 3 good
**Presentation:** 2 fair
**Contribution:** 3 good
**Rating:** 5
**Confidence:** 4

**Summary:**

This paper present an empirical defense against adversarial (evasion) attacks on image classifiers. The authors build their defense on the premise of improving the overfitting issue experienced during adversarial training. They propose that this overfitting occurs because high confidence adversarial examples are added in the training data of the model during adversarial training. They propose a metric to measure the confidence level of adversarial examples and develop a training method to force the model to learn weights that yield low confidence adversarial examples. By doing so, they reduce the generalization gap in robustness obtained through adversarial training (and other similar methods). Models trained using the proposed method exhibit higher clean and adversarial accuracy than baselines.

**Strengths:**

1. The authors provide novel evidence with regards to the correlation between confidence on adversarial examples and robustness generalization gap: the higher the confidence, the bigger the generalization gap.
2. Using the above insight, authors devise a novel method for improving the robustness generalization gap. Their method measure the variance of the output distribution across all labels and aims to reduce it during adversarial training.
3. The proposed method can be combined with any defense method that makes use of the strategy of training on adversarial examples.
4. The design choice of the proposed method is adequately justified. While reading the paper, one of the things that came to my mind was that the method might be better implemented as a regularization term. I appreciate that the authors compared this obvious design choice with their chosen design, and demonstrated that it works better.

**Weaknesses:**

1. The writing quality of the paper can be improved. It was hard to read through the paper and grasp the information being conveyed, majorly because of awkwardly worded or vague sentences. I highly recommend authors to undertake a major re-writing of the paper and convey the important insights as well as method design discussions in a more articulate manner.

2. The jump from label-level variance metric to adversarial certainity metric is abrupt. The motivation for the adversarial certainity metric is lacking. Ease of optimization is understandable, however it is unclear why one would compute variance using logits instead of softmax outputs.

3. Potential issue with reproducing results using TRADES. The AutoAttack paper reports robustness against AA as 53.08% (for the last checkpoint made publicly available by TRADES authors) while in this paper this number is reported as 46.53%. Therefore, either the model was incorrectly trained or AA wasn't correctly executed. It is important to figure out the source of this discerpancy.

4. No discussion regarding computational overhead of the proposed method. It appears that the proposed method doubles the number of times adversarial examples must be generated during training. Since the generation of adversarial examples is the bottleneck process in adversarial training, it appears that the proposed method comes at a huge cost to computational efficiency. One must ask then, whether the improvements in clean and robust accuracy are worth it? From a research perspective, the results may be intersting but the proposed method seems to be infeasible in real world settings.

5. No discussion regarding adaptive attacks. An adaptive adversary would try to generate low confidence adversarial inputs by including both adversarial certainity and cross-entropy loss in the attack objective. Such an attack has not been discussed in the paper at all. Another effective strategy for an adaptive attack would be to find an adversarial example that matches the latent space feature representations (eg, penultimate layer) of an input from another class (see [a] for more details). Overall, I suspect that because the proposed method 'flattens' the output space, the gradient based attacks might be failing to be effective.

6. The authors haven't taken adequate steps to present the improvements introduced by their method as statistically significant figures. This would entail running PGD/CW/AutoAttack with multiple random restarts (20 to 50), or running the same attack multiple times (using different random inits) and reporting the mean and 95% confidence interval across all runs.

7. Comparison with closely related prior work (Setlur et al., 2022) is missing. The method proposed by this prior work and the one proposed in this paper are based on the same premise: perform regularization (or, reduce overfitting) by reducing confidence on (or, along the direction of) adversarial examples. Both these methods can be combined with prior adversarial training based methods. Therefore, it is very important to compare the proposed method with this prior work, across a range of adversarial training methods.

**Writing Issues**
1. Distillation is not a secure defense, hence listing it as one (in intro, para 2, line 2) is improper.
2. In intro, 2nd para, authors list several (dissimilar) defenses and then say that adversarial training is the most effective among them all. This statement is not technically sound. For example, randomized smoothing is a fundamentally different defense than adversarial training in terms of the nature of robustness conferred (certified vs. empirical robust). As such, the robustness of models trained using these two methods can not be directly compared.
3. Ambiguous statement in intro, 2nd para "sota adversarial training methods can not achieve satisfactory robustness performance", what does "satisfactory" mean here?
4. Typo fig 2(a), y axis label: "Label-Lavel" => "Label-Level"

**References**

[a] Sabour, S., Cao, Y., Faghri, F., and Fleet, D. J. Adversarial manipulation of deep representations. International Conference on Learning Representations, 2016.

**Questions:**

1. The AutoAttack paper has evaluated prior defenses using publicly available checkpoints. In case of TRADES, they have used the WRN34 checkpoint at end of 100 epochs (last checkpoint). For this checkpoint, they report accuracy of 53.08% against AA. However, for the same model/checkpoint, authors report 46.53%. What is the cause of this discrepancy?

2. Seems like the proposed method has poor computational efficiency as compared to adversarial training (which already has very poor computational efficiency). Luckily, there are methods that significantly imporve the computational efficiency of adversarial training [b,c]. Based on the content of the paper, it appears that the proposed method should work in conjunction with these methods too. But are there any results to indicate how effective the combination will be? I highly recommend adding these results (even if in appendix) in a future iteration of the paper.

3. What happens if the the adversary tries to generate an adversarial input that maintains low confidence while trying to fool the classifier, i.e., include both adversarial certainity and cross entropy loss in the attack objective. This would be the most straight forward adaptive attack.

**References**

[b] Wong, Eric, Leslie Rice, and J. Zico Kolter. "Fast is better than free: Revisiting adversarial training." International Conference on Learning Representations. 2019.

[c] Shafahi, Ali, et al. "Adversarial training for free!." Advances in Neural Information Processing Systems 32 (2019).

---

### Official Review · Reviewer_v7sc · 2023-10-31

**Soundness:** 2 fair
**Presentation:** 3 good
**Contribution:** 2 fair
**Rating:** 5
**Confidence:** 3

**Summary:**

This work finds that models exhibit superior robust generalization performance have more even predicted label distributions. Then, they introduce an adversarial training method that uses less certain examples by reducing adversarial certainty. They provide some experimental results to verify the efficacy of their approach in addressing robust overfitting problem.

**Strengths:**

- Easy to follow, clear presentation.
- Numerous empirical results.

**Weaknesses:**

- The analysis about even label distribution is limited, e.g., this work does not provide some theoretical analyses.
- There are numerous experiments, however, I cannot find some impressive empirical results, especially under auto attack. ref: https://arxiv.org/abs/2302.04638; https://arxiv.org/abs/2202.10103; https://arxiv.org/abs/2004.05884
- All in all, I think this is a borderline paper, it provides a story but the empirical results and the motivation analysis are limited.

**Questions:**

See weaknesses.

---

### Official Review · Reviewer_whj7 · 2023-11-06

**Soundness:** 2 fair
**Presentation:** 3 good
**Contribution:** 3 good
**Rating:** 5
**Confidence:** 4

**Summary:**

This paper studies on the robust overfitting phenomenon. Specifically, adversarially trained models overfit to the training attacks and fail to generalize during testing. The authors reproduce this phenomenon and propose a novel metric called "Adversarial Certainty" to measure how severe the overfitting / overconfidence is. The authors claim that minimizing Adversarial Certainty helps the model generalization and introduce an extra-gradient-like algorithm to minimize Adversarial Certainty during training. Finally, experiments show the proposed method outperforms state-of-the-art. Adversarial Certainty metric is also verified consistent with the claim. In general, this paper is straightforward and easy to follow.

**Strengths:**

* The introduction of "Adversarial Certainty" is novel. The authors conduct through experiments to show the correlation between "Adversarial Certainty" and "generalization at testing".

* The idea of minimizing "Adversarial Certainty" is interesting and looks promising given the extensive experiments on CIFAR-10.

**Weaknesses:**

**Technical:**

The main weakness of this paper is that the algorithm and the following discussions are based on intuitive explanations and are lack of solid support. The authors do provide some support but I am afraid that they are not convincing enough. Specifically there are two questions:
* Does lower *Adversarial Certainty* imply better robust generalization?
* Does the proposed algorithm indeed solves the problem (2)? Does it have anything to do with extra-gradient?

Regarding the first question, the authors provide experimental evidence and intuitive justifications. I think this argument my hold after a long period of training but may not be true at any time. The reason is that the definition of *Adversarial Certainty* does not have label $y$ involves but the definition of *robust generalization* has. In other words, think of the extreme case in which we have a random classifier. The *Adversarial Certainty* is very low but *robust generalization* is low as well. Furthermore, Figure 3 (b) shows the trends that when *Adversarial Certainty* gets very small, *robust generalization* starts to decrease. Think of the case at the very beginning of training in which the *Adversarial Certainty* is not very large. Minimizing the *Adversarial Certainty* at the moment might not be beneficial.

Regarding the second question, I am not convinced that proposed algorithm (3) solves problem (2). I understand that (2) is very difficult to solve due to its non-convex and non-concave nature. However even under the sense of approximation, I am not sure how (3) approximates  (2) or whether (3) indeed minimizes *Adversarial Certainty*. The following discussions are intuition based and are not rigorous. For example, the authors claim that *Adversarial Certainty* is decrease after the first step. This is reasonable but how about the second step? How can we show that *Adversarial Certainty* is not increased during the section step? Another question is that, how (3) is related to extra gradient? Both steps in (3) use different objective functions and the second step update is based on $\theta_{t+0.5}$.

**Some other issues:**
* In the last paragraph of page 7, I guess I may not understand the setup. Why not just run AT and AT-EDAC and report *Adversarial Certainty* at each epoch?
* The study of *Alternative Regularization Method* on page 8 is actually very interesting. What is the exact formulation of the regularization term? It seems that the performance gap between EDAC and EDAC_reg is marginal, which strengthens my concern of (3). Then why not simply use regularization?

**Writing:**

Although the paper in general is straightforward and easy to follow, I believe the presentation can be greatly improved.
* There are a bunch of missing definitions, e.g. AT on page 5, $L_\mbox{rob}$ on page 5. They are actually defined later but make the reader confusing when they first appear without being defined.
* In paragraph 2, the relegation of definitions make it difficult to follow. Important definitions such as *robust generalization* should remain in the body.
* There are lots of unnecessarily long sentences making it difficult to read, e.g. first sentence in the second paragraph of section 2 is four-line long.

**Summary**
* The idea that minimizing *Adversarial Certainty* is interesting, novel and seems promising. The motivation of two steps update of (3) is not  well explained. The relation between (3) and extra gradient in not clear. Given the similar performance of EDAC_reg, the benefit of (3) is unknown. The presentation can be improved by clearing up the definitions and making the explanations succinct.

**Questions:**

Please see above section.

---

### Official Review · Reviewer_QZ4H · 2023-11-06

**Soundness:** 2 fair
**Presentation:** 3 good
**Contribution:** 2 fair
**Rating:** 3
**Confidence:** 4

**Summary:**

The paper studies the concept of robust overfitting, where DNNs overfit when trained on adversarial data, after an initial increase in adversarial robustness. The concept was first found in Rice et al., 2020 - and this paper tries to mitigate it by understanding it through the lens of adversarial uncertainty - i.e the probability of the model on adversarial inputs. They find that the model is overconfident on adversarial inputs at training time, but not on those at test time - which indicates miscalibration of the underlying adversarially trained model. This is then mitigated by adding an extra gradient step to generate inputs with low adversarial certainty during training time to avoid overfitting.

**Strengths:**

The hypothesis that decreasing the gap in certainty of train vs test-time adversarial examples is tested empirically, and across different regularization and adversarial training baselines.

**Weaknesses:**

* The choice of step size is done by optimizing for test-time robustness, and not on robustness metrics on a validation split. Optimizing hyper-parameters based on test-time metrics is not acceptable in ML model selection, and a validation split evaluation is warranted. This also brings into question the soundness of results presented in Tables 1-3, as the step size impacts the test-time robustness quite a bit per Fig 3b
* Constrained search space of the initial parameter \theta’ is not well justified as to what the search space is minimizing - there seems to be a trade-off between avoiding overfitting, and optimizing for low Adversarial certainty - which is not validated -but only an intuition is provided based on correlation analysis between train and test time measures.
* Confidence intervals in the Tables are not presented - which makes it harder to understand if these results are statistically significant. For example, in the results section, the claim of “In general, the improvement of EDAC on AWP and Consistency is not as significant as previous ones” cannot be verified.

If the above points around soundness are clarified, I'm willing to increase the score.

**Questions:**

* Notations L, B in Eqn 2, L_{rob} in Eqn 3 are not defined.
* Numbers in brackets in Table 1 are not explained
* Figure 5 is missing EDAC versions of the methods - would be helpful to see how the proposed method helps.

---

### Official Review · Reviewer_aLt2 · 2023-11-09

**Soundness:** 2 fair
**Presentation:** 2 fair
**Contribution:** 2 fair
**Rating:** 3
**Confidence:** 4

**Summary:**

GOAL:
Investigate the “robust overfitting phenomenon”
After Adversarial Training, the new robust model outputs a more flat distribution of labels compared to non-robust models. It overfits the difficult adversarial data at the expense of the in distribution data.

METHOD:
add an  extragradient step in the adversarial training framework to search for models that can generate adversarially perturbed inputs with lower certainty

Contributions:
1. Observe that the label distributions predicted on train time adversarial examples are:
- more even (flat?) for “models with better robust generalization ability”
- over confident in predicting labels for models produced by adversarial training

2. Introduce the concept, “adversarial certainty”:
- Defined in Eq 1, as what’s the variance in the final logit layer of a NN, when we do an epsilon bounded attack on an input.
- Describe how does this concept relate to other concepts around robustness (4.2) such as AWP. The main difference between those two is that EDAC contains a half step where the model params are updated to make it easier to reduce adversarial certainty.

3. Propose EDAC method for adversarial training. Procedure:
- Minimize (delta adversarial certainty / delta model weights) by a two step optimization: first adjust NN weights to decrease adversarial certainty on the training set, and then the second step tries to do a typical loss where you try to still pick the correct answer.
- They generated adversarial examples during training that are less certain
- These adversarial examples with low certainty are used to optimize robustness
- End: Generalization gap between train and test sets is lower

**Strengths:**

1. EDAC method can be combined with other adversarial training methods.
2. Figure 1 provides an quick and easy intuitive understanding of the concept. I appreciate the use of the heat maps.

**Weaknesses:**

In this section, I'll first justify my overall negative rating and then list out specific weaknesses.

OVERALL:
I am not sure that the adversarial robustness angle to the problem changes the core problem of overfitting to training data, and thus not doing so well on the test data. Some of the stated contributions (like showing that high adversarial uncertainty led to better test set performance, or the correlation of the two) were not fully achieved, and have been achieved in the past AWP paper. (Based on my understanding, I'm arguing that AWP's notion of flatness does not materially differ from "adversarial certainty" introduced in this paper). Finally, the empirical results do not hold up: EDAC is not better than AWP, and a clear head to head comparison is missing. It's also not clear from Table 3, that EDAC marginally improves AWP performance (you would need confidence intervals because the improvements are so slim).


SPECIFIC WEAKNESSES:
1. A contribution listed was to show that flatness in the logit distribution leads to better generalization. But that’s been shown before in the AWP paper you cite. I’m still not very clear on the difference between AWP and this method: Is it that you generate the adversarial examples after that intermediate step of adjusting the model parameters to themselves increase uncertainty?

2. If I’m understanding the Section 2, subheading, “Our perspective.” you’re arguing that the larger gap between train and test performance on a given adversarial attack limits adversarial robust generalization -- which is defined as the performance against that adversarial attack on a held out set of data. That is a tautology.

3. Also in the same section: I think what you’re trying to get at is to state your hypothesis of  “because the model cannot generate perturbed training inputs with low enough certainty, which is different from the typical generated adversarial examples during testing time.” But I’m also not really sure what that sentence means. I take it that there are examples that are “more certain” (less logit variance) and “less certain,” (more logit variance), and we do worse on the test when it is all “less certain” since we haven’t seen it yet, so we need to explore “less certain examples?” Is that just arguing for active learning by generating more data from a given attack (i.e if we get more examples of the attack where we are right now a bit confused on the label, then we will do better on the test set?”). Because that makes sense, but feels a bit obvious. Further, I don’t understand what the adversarial attack has to do with it -- it would be true of any data generation process. In which case, I don’t know that it’s a significant advancement from other papers that use extra gradient steps? ( I guess, what about the adversarial robustness makes this problem different from any overfitting/generalization problem?)

4. Figure 3 doesn’t show a correlation. I’d want a plot that plots adversarial certainty against robustness accuracy (with and without EDAC) -- or a summary metric. I don’t really see that information presented for the  “without EDAC”. Also, you don’t discuss the trend in 3b where the orange line of test set performance is going down as adversarial certainty further decreases. (In fact the AWP paper which tries to make a very similar claim doesn't claim outright correlatio, noting: "Therefore, a flatter weight loss landscape does directly lead to a smaller robust generalization gap but is only beneficial to the final test robustness on condition that the training process is sufficient (i.e., training robustness is high)")

5. Table 3: The improvements are very slim (esp. against AWP), such that I’m not sure what’s a significant difference. Confidence intervals needed to conclude that EDAC does have a positive cumulative effect.

6. Table 2 and 3: In fact when you compare row 2 of Table 2 (EDAC only) vs row 1 of Table 3 (AWP only), AWP does very significantly better. (As an aside, please include these head to head comparisons of different methods more clearly).

7. The actual sentence writing is quite hard to parse, and significantly hurts my understanding of the work.
 ex. This sentence in the abstract should be multiple sentences.
“...t overconfident models produced during adversarial training could be a potential cause, supported by the empirical observation that the predicted labels of adversarial examples generated by models with better robust generalization ability tend to have significantly more even distributions. ”
OR
“EDAC first finds the steepest descent direction of model weights to decrease adversarial certainty, aiming to generate less certain adversarial examples during training, then the newly generated adversarial examples with lower certainty are used to optimize model robustness.”
OR
“Besides, we investigate the cooperation of adversarial certainty with other helpful insights for model robustness, where our method shows further generalizability by compatibly improving them (Section 4.2).”

**Questions:**

1. What about the *adversarial robustness* makes this problem different from any overfitting/generalization problem that uses extra gradient steps/awp?


2. What is the criteria for selecting “best model” in Fig 1? Is it the one with the best test set performance?
3. In Figure 2, which is on the train set as well, how does the best model have *lower* label variance? I thought the Best model is the one with less confidence in the label (Fig 1c) and should therefore be more varied in the labels it predicts on the train set?
4. Eq 2: How do you know what C(\theta) is? What defines a feasible search region? ex: Is this some meta parameter you use to enforce normed distance?
5. Eq 2: What is B_{epsilon}(x) on the left side under tha max? You don’t define it I think, but for now I’m presuming that’s the perturbation for x? Is it the attack?

**Details Of Ethics Concerns:**

No ethical concerns that are not relevant to all generalization studies.